# G Protein-Coupled Receptor–Ligand Pose and Functional Class Prediction

**DOI:** 10.3390/ijms25136876

**Published:** 2024-06-22

**Authors:** Gregory L. Szwabowski, Makenzie Griffing, Elijah J. Mugabe, Daniel O’Malley, Lindsey N. Baker, Daniel L. Baker, Abby L. Parrill

**Affiliations:** Department of Chemistry, University of Memphis, Memphis, TN 38152, USA; gszwabowski@gmail.com (G.L.S.); makenziegriffing@outlook.com (M.G.); elm4026@med.cornell.edu (E.J.M.); domalley@memphis.edu (D.O.); lindsey.baker@yale.edu (L.N.B.)

**Keywords:** G protein-coupled receptor (GPCR), docking, random forest classifier, interaction fingerprint, machine learning

## Abstract

G protein-coupled receptor (GPCR) transmembrane protein family members play essential roles in physiology. Numerous pharmaceuticals target GPCRs, and many drug discovery programs utilize virtual screening (VS) against GPCR targets. Improvements in the accuracy of predicting new molecules that bind to and either activate or inhibit GPCR function would accelerate such drug discovery programs. This work addresses two significant research questions. First, do ligand interaction fingerprints provide a substantial advantage over automated methods of binding site selection for classical docking? Second, can the functional status of prospective screening candidates be predicted from ligand interaction fingerprints using a random forest classifier? Ligand interaction fingerprints were found to offer modest advantages in sampling accurate poses, but no substantial advantage in the final set of top-ranked poses after scoring, and, thus, were not used in the generation of the ligand–receptor complexes used to train and test the random forest classifier. A binary classifier which treated agonists, antagonists, and inverse agonists as active and all other ligands as inactive proved highly effective in ligand function prediction in an external test set of GPR31 and TAAR2 candidate ligands with a hit rate of 82.6% actual actives within the set of predicted actives.

## 1. Introduction

G protein-coupled receptors (GPCRs) constitute one of the largest protein superfamilies in the human genome, encompassing over 800 members [1]. These receptors act to relay extracellular signals to their intracellular effectors in many cellular signaling pathways and, thus, play a critical role in several aspects of human physiology. Consequently, dysregulation of GPCR signaling can lead to diseases, such as diabetes, cancer, and nervous system disorders [2]. Given that these receptors are of immense physiological importance, GPCRs currently serve as targets for ~30% of FDA-approved drugs [3] and remain the center of focus for the development of many novel therapeutics. However, FDA-approved drugs only target 108 of the 360 known “druggable” non-olfactory GPCRs [4], indicating that a substantial swath of GPCR targets are yet to be clinically leveraged. Furthermore, ~25% of GPCR are still classified as “orphans”, signaling that they lack a known endogenous ligand and often possess an ambiguous or undefined physiological role [5]. For these understudied GPCRs, the ability to better understand and modulate their physiological roles may illuminate poorly characterized pathways relevant to the development, progression, and treatment of disease. Thus, the identification of novel ligands for understudied GPCR targets is a critical first step in the development of novel therapeutics.

Recently, GPCR ligand identification studies have employed virtual screening (VS) [6,7,8], which allows for the computational screening of large compound libraries to prioritize sets of compounds for more targeted in vitro screens. Among the various techniques used in VS studies, molecular docking remains one of the most prevalent methods of candidate prioritization due to its low financial and computational costs [9]. Molecular docking aims to accurately predict a ligand’s binding mode within the constraints of a protein target’s binding site [10]. During the docking process, potential ligand binding modes are first algorithmically sampled and then quantified and ranked with a scoring function [11]. These scoring functions are often used to identify docked poses that may reflect a ligand’s true binding mode [12], although the pose most similar to the true binding mode may not be ranked as the most energetically favorable [13]. Of even greater concern for virtual screening is the fact that multiple studies have found that pose scoring functions poorly correlate with binding affinity [14,15]. Additionally, virtual screening results in several class A GPCR examples improved with target-specific reweighting of the energetic contributions in the scoring function, suggesting imperfect transferability of scoring functions between different protein targets, even within the same family [16]. Furthermore, since classical scoring functions primarily use simplified mathematical approximations of protein–ligand binding interaction terms to estimate enthalpic contributions to ligand binding [11], they do not provide a method of segregating active compounds from inactive compounds [14], and nor do they distinguish between various functions of active compounds (agonists, antagonists, inverse agonists) [17]. Recent applications of deep learning methods to the docking problem have thus far failed to outperform classical docking methods due to insufficient consideration of intra- and intermolecular validity with common errors in bond lengths, atomic geometry, stereochemistry, and interatomic distances that violate underlying physics [18]. Thus, alternative methods of (A) separating active compounds from inactive compounds and (B) determining the functional specificity of active compounds during a VS workflow are necessary. Recent applications of machine learning to identify GPCR ligands may prove useful in eliminating screening candidates that are unlikely to interact with any GPCR [19,20]. However, these tools fail to predict which member of the GPCR family will be the target of molecules identified as likely to target GPCRs. Thus, alternative approaches are still of high value.

Fortunately, previous efforts to relate GPCR structure to function via the comparative analysis of binding site composition and types/strengths of ligand–receptor contacts observed in GPCR–ligand complexes representing differing activation states provide a basis from which to functionally characterize in silico GPCR–ligand pairings. For example, comparative structural analyses of GPCRs using residue contact maps by Hauser et al. [21] and Venkatakrishnan et al. [22] related differences in the strengths and types of GPCR residue contacts to observed variations in GPCR function. Altogether, the results observed in these (and other) studies lend themselves to potential improvements in GPCR–ligand pose prediction and the development of a machine learning classifier that uses fingerprints consisting of the types, strengths, and locations of ligand–residue contacts observed in the resulting in silico GPCR–ligand complexes to predict whether a ligand is likely to bind to a GPCR.

In this work, we first utilized ligand–receptor complexes retrieved from the Protein Data Bank (PDB) [23] to develop several ligand interaction fingerprints to assess as docking site selection tools to determine whether such fingerprints improve pose prediction based on the expectation that pose quality would impact the development of the subsequent machine learning classifier. Fingerprints generally improved the percentage of ligand–receptor pairs with successful sampling outcomes but had minimal impact on success of scoring outcomes. We thus concluded that there was inadequate benefit for fingerprint-based site selection over automated site selection for the development of a machine learning classifier to predict ligand binding and function, which would use a ligand–receptor complex based on the top-ranking pose as the input.

We then detail the development of a classifier that aimed to predict whether a ligand is likely to orthosterically bind to a GPCR (binders and nonbinders are herein referred to as actives and inactives, respectively) based on its ligand–receptor complex. In addition, we investigated the classifier’s ability to distinguish between different types of ligand function (agonists, antagonists, inverse agonists) for ligands predicted to be active. At the heart of this classifier is the random forest algorithm, an ensemble learning approach that constructs many decision trees and outputs predictions based on the predicted class votes of the individual decision trees [24]. To train and test the classifier, an internal dataset containing interaction profiles (representing the categorical interaction types of Hbond: hydrogen bonding, Metal: metal–complexed ionic interactions, Ionic: ionic interactions not involving a metal, Arene: pi–pi or cation–pi interactions, or Distance: van der Waals interactions and numeric interaction energies observed at each residue comprising a GPCR) extracted from 1820 experimentally determined and docked ligand–receptor complexes of known active and inactive GPCR ligands was constructed. To retrieve inactive ligands for GPCR targets, the Database of Useful Decoys: Enhanced (DUD-E) [25] was used since its repositories possess lists of inactive and decoy ligands for multiple GPCR targets. With this dataset, the use of ligand–receptor complexes retrieved from the PDB [23] in combination with ligand–receptor complexes generated via molecular docking with experimentally determined or modeled structures allowed for the classifier to be trained on biologically observed ligand–protein interactions as well as interactions observed in docked poses, both of which are frequently encountered in GPCR–ligand identification studies. Furthermore, considering that only 140 of the over 800 known GPCRs possess published structures on the PDB as of 21 February 2023 [23,26], we found it necessary to use modeled structures to generate a portion of the docked ligand–receptor complexes in the dataset to allow for the classifier to be applicable to cases where a GPCR target does not possess an experimentally determined structure.

To emulate the application of our classifier to a virtual screening workflow, two orphan GPCRs with putative endogenous ligands were selected as targets for which to construct a dataset for external validation (herein referred to as the external dataset), namely G protein-coupled receptor 31 (GPR31) and trace amine-associated receptor 2 (TAAR2). Considering that neither of these targets possess published experimentally determined structures or large sets of known active ligands, we found them to be adequate for external validation of the classifier. Using ligands shown to be active or inactive for each target in prior studies by Guo et al. [27] (GPR31) and Borowsky et al. [28] (TAAR2), an external dataset containing interaction profiles for ligands docked into modeled structures of GPR31 and TAAR2 was constructed and used to further assess our classifier’s accuracy in predicting ligand function based on a given binding mode. In addition to simply allowing for the assessment of whether the classifier could correctly predict the binary activity class (active or inactive) of known active and inactive ligands for these targets, the use of modeled structures from various sources (in-house homology models [29,30,31], GPCRdb [26], AlphaFold [32,33]) during construction of the external dataset ensured that the classifier’s performance was assessed in the context of structurally variable GPCR models.

Overall, this work demonstrated the ability of our random forest classifier to accurately predict the binary activity class of GPCR ligands in complex with experimentally determined structures (with 97.6% of ligands classified as actives being true actives) and more importantly, in complex with modeled structures (with 80.6% of ligands classified as actives being true actives). Furthermore, classification of our external dataset resulted in 82.6% of the GPR31/TAAR2 ligands classified as actives being true actives. Although prioritization of compounds for experimental in vitro screens against GPCR targets remains a challenge, we hope that the success demonstrated by this classifier will assist in GPCR ligand identification efforts.

## 2. Results and Discussion

With this work, we aimed to address two significant research questions. First, do ligand interaction fingerprints provide a substantial advantage over automated methods of binding site selection for classical docking? Second, can the functional status of prospective screening candidates be predicted from ligand interaction fingerprints using a random forest classifier?

In the first portion of our results (Section 2.1), we discuss the development and assessment of ligand interaction fingerprints for binding site selection to address the first research question. In the remaining sections, we address the second research question using the six types of ligand–receptor complexes shown in Figure 1 as the internal dataset for random forest classifier training. In Section 2.2, we discuss the construction and retrieval of modeled GPCR structures used to represent cases where a target does not possess an experimentally determined structure (required to generate ligand–receptor complexes of type 5 and 6 in Figure 1). In Section 2.3, we detail the development of the internal dataset illustrated in Figure 1 containing experimentally determined and docked GPCR–ligand complexes used to train and test the random forest classifier. In Section 2.4, the preparation of the dataset used to externally validate the machine learning classifier (containing ligand–receptor complexes of active and inactive ligands for two understudied GPCR in complex with modeled structures from various sources) is discussed. The subsequent section (Section 2.5) describes the extraction of interaction profiles serving as feature sets that were used to train the random forest classifier. Lastly, Section 2.6 details a performance assessment of the random forest classifier when predicting the activity and functional specificity of ligands represented in our testing and external datasets.

### 2.1. Ligand Interaction Fingerprint Development and Assessment

Global and activation state-specific ligand interaction fingerprints were developed as potential tools to define GPCR docking sites in the hope that knowledge-based docking site definition would produce more accurate poses as input for a machine learning classifier. Global fingerprints were derived using ligand interaction patterns observed in a large set of experimentally-determined structures that included receptors in the active, intermediate, and inactive states. Two global fingerprints were developed, one with and one without a ligand interaction score threshold. Three receptor state-specific fingerprints were derived from interaction patterns observed in subsets of structures based on common receptor activation states (active, inactive, intermediate). The use of fingerprints was compared with the automated SiteFinder function (a geometric method based on Alpha Shapes [34]) in MOE [35] for docking site selection, with pose quality assessed using the root-mean-square deviation (RMSD) of the atomic position relative to the reference complex as the pose quality measure.

The RING Server [36] was used to identify GPCR positions interacting with bound ligands in experimental GPCR complexes [37,38,39,40,41,42,43,44,45,46,47,48,49,50,51,52,53,54,55,56,57,58,59,60,61,62,63,64,65,66,67,68,69,70,71,72,73,74,75,76,77,78,79,80,81,82,83,84,85,86,87,88,89,90,91,92,93,94,95,96,97,98,99,100,101,102,103,104,105,106,107,108,109,110,111,112,113,114,115,116,117,118,119,120,121,122,123,124,125,126,127,128,129,130,131,132,133,134,135,136,137,138,139,140,141,142,143,144,145,146,147,148,149,150,151,152,153,154,155,156,157,158,159,160,161,162,163,164,165,166,167,168,169,170,171,172,173,174,175,176,177,178,179,180,181,182,183,184,185,186,187,188] downloaded from the PDB [23] described in Appendix A. These interactions were used as a basis to develop ligand interaction fingerprints. As an example, Figure 2 displays interaction sites observed in two opioid κ receptor complexes, one with an active receptor state bound with an agonist, and the other with an inactive receptor state bound with an antagonist. Some interactions were observed universally in both complexes, while others were dependent on the receptor activation state and bound ligand. The opioid κ receptor in the active state showed a greater number of ligand interaction sites in the third and fourth transmembrane (TM) segments, whereas the opioid κ receptor in the inactive state showed a greater number of ligand interaction sites in the TM1, TM6 and TM7 segments. Broadening this type of analysis to identify patterns across multiple receptor–ligand complexes, we developed both global fingerprints using the combined set of GPCR–ligand complexes in Appendix A and fingerprints using subsets of structures with the same activation state.

Considering that class A GPCR sequence lengths differs from receptor to receptor (with a majority of GPCRs possessing sequence lengths from 310 to 470 residues [189]), the Ballesteros–Weinstein (BW) numbering scheme [190] was used to ensure consistent indexing of transmembrane residue positions across all ligand–receptor complexes in our datasets. This numbering system uses a pattern X.YY. Here, X is either the single number of the TM helix in which a residue is found or the two numbers of the TM helices on either side of the loop in which the residue is found. On the other hand, YY is a number that indicates a position relative to the most conserved residue in the helix or loop which is assigned position 50. The percentage of receptors exhibiting an interaction at a given site was weighted to ensure that each of the 60 receptors represented among the 311 structures in the dataset had an equal weight in determining the interaction percentage. Appendix A shows the weighted interaction percentages as a function of common geometric locations across the full set of 311 structures. A similar set of weighted interaction percentages was generated using only those interactions assigned an interaction score of 0.5 [30]. This score excludes interactions with distances greater than 3.93 Å. Sets of high-scoring weighted interaction percentages were also generated using only structures with receptors in active, inactive, or intermediate states. The fingerprints of 10–15 sites with the highest interaction percentage generated from each of these weighted interaction percentages are shown in Table 1. The selection of 10–15 sites for each fingerprint required the use of different weighted interaction percentages to select sites, within a range from 35% to 60%. None of the interaction fingerprints included any sites from TM1 or TM5. Only global fingerprint A (with no interaction score cutoff) and the intermediate fingerprint included any sites from TM2. All fingerprints shared sites from TM3 (3.29 and 3.33) and TM 6 (6.55). The active and inactive fingerprints shared 9 out of 10 sites, differing only in the inclusion of 7.35 in the active fingerprint and 6.52 in the inactive fingerprint.

The performance of interaction fingerprints was compared with the automated SiteFinder function to guide docking site selection in the MOE software [35]. Receptor and ligand structure sources for each docking calculation, as well as the fingerprints tested, are shown in Appendix A [37,38,39,40,41,42,43,46,47,48,49,50,52,59,81,85,88,91,100,102,121,122,123,124,125,148,149,161,162,175,176,191,192]. The percentage of docking calculations that produced ligand poses with RMSD values in the successful (<2 Å), acceptable (2–3 Å), successful + acceptable (<3 Å), and unsuccessful (>3 Å) categories were used to compare fingerprint performance to the automated SiteFinder performance. The lowest RMSD pose within the 400 poses generated in each docking run represented the quality of pose sampling. The lowest RMSD pose ranked in the top 5 poses (lowest energy) represented the quality of scoring. Appendix A and Table 2 show that a modest improvement in the percentage of docking calculations sampling successful poses when guided by global fingerprints (35.0%) compared to automated guidance (30.0%). Successful poses were least frequently found in the top 5 when docking sites were selected using fingerprint A at only 10.0% of docking calculations. Automated site selection produced successful top-ranked poses in 15.0% of docking calculations. Site selection based on fingerprint B, which emphasized closer contacts, showed a modest improvement, with 20.0% of docking runs showing successful results among the top 5 scored poses.

Appendix A and Table 3 show that activation-state specific fingerprints provided modest decreases in the percentage of unsuccessfully sampled poses for the inactive and intermediate fingerprints (25%) compared with automated site selection (37.5% and 31.3% unsuccessful for inactive and intermediate, respectively). However, these modest improvements did not reduce the unsuccessful outcomes when considering only the top 5 poses.

Neither global nor activation state-specific fingerprints provided a substantial benefit over automated docking site selection with regards to the quality of docked poses scored within the top five. Classical docking using automated site selection was, therefore, used for the development of a machine learning classifier to predict ligand binding and function.

### 2.2. Protein Modeling

The internal dataset used to develop our random forest classifier (Figure 1, ligand–receptor complex types 5 and 6) required modeled structures for the 5 DUD-E GPCR targets (AA2AR, ADRB1, ADRB2, CXCR4, and DRD3). For each DUD-E GPCR target, a set of three homology models representing “best” and “normal” cases for template selection were constructed using our benchmarked homology modeling protocol (Table 4) [29,30,31]. We chose to develop multiple homology models for each target (rather than just a single homology model) to reflect the range of information that may or may not be available for any GPCR target at the center of a ligand identification study. For example, the pair of best-case homology models constructed for each target represents a situation in which active and inactive state structures of a closely related receptor that binds the same endogenous ligand as the target GPCR are available for use as template structures. In contrast, normal-case homology models represent a more often encountered situation in GPCR–ligand identification studies, where an understudied GPCR is simply modeled using a template structure selected with a metric measuring intra-GPCR similarity (in this case the CoINPocket score [30]) available for a target at that time. Since endogenous ligands may be unknown when constructing a normal-case homology model (such as with orphan GPCRs), template structures selected for the construction of normal-case homology models may not bind the same endogenous ligand as the GPCR being modeled. When generating homology models for a GPCR, the use of a template structure that is more closely related (and, thus, more similar in terms of amino acid sequence) to a target GPCR is thought to lead to homology models that more closely reflect experimentally determined structures of a GPCR. Thus, we hypothesized that the development of separate homology models with closely related template structures (best-case homology models) and more distantly related template structures (normal-case homology models) would result in homology models that reflect a range of structural accuracy. After the construction and loop refinement of each target’s best- and normal-case homology models, RMSD values were calculated using an alpha carbon superposition of each homology model onto the lowest resolution experimentally determined structure that matched the homology model template’s activation state (Table 4). When RMSD values were calculated, active and inactive state reference structures were available for all targets except CXCR4, which only possessed inactive state experimentally determined structures. RMSD values ranged from 2.71–5.68 Å for best-case homology models and 2.76–5.47 Å for normal-case homology models. On average, best- and normal-case homology models exhibited RMSD values of 3.96 Å and 4.07 Å, respectively. In our hands, this indicated that the consideration of whether a template structure possessed the same endogenous ligand as the GPCR being modeled did not lead to a stark difference in homology model quality. However, observed RMSD values support the use of the CoINPocket similarity score as a metric for homology model template selection regardless of whether a template structure binding to the same endogenous ligand as the target GPCR being modeled is available.

For the external dataset, four modeled structures were generated or retrieved for each target: one was constructed with our in-house GPCR modeling protocol [29,30,31], two were retrieved from GPCRdb [26], and one was retrieved from the AlphaFold Protein Structure Database [33]. When generating the in-house homology model for each target in the external dataset, template structures were selected using the CoINPocket similarity metric [30], and loop modeling was again performed after the determination of loop anchor residues (Appendix A). The two GPCRdb homology models retrieved for each external set target were, respectively, constructed with active and inactive state template structures using the GPCRdb homology modeling pipeline [201].

### 2.3. Internal Dataset Preparation

In virtual screening workflows concerning GPCR, structures used to probe receptor function can originate from a variety of sources. For example, a ligand identification campaign for a well-studied GPCR (such as ADRB2 [3]) is likely to rely on a plethora of experimentally determined structures that reflect multiple activation states of the GPCR in complex with functionally diverse ligands. In contrast, efforts to identify novel ligands for understudied GPCRs (such as orphan GPCRs [202]) are more likely to use modeled structures due to the lack of an experimentally determined structure. As such, the internal dataset used to develop the random forest classifier in this work contained 1820 class A GPCR–ligand complexes of mixed origin that were grouped into six categories that reflect the different types of structures that can be encountered during a GPCR ligand discovery workflow (Figure 2). The first structural category within the internal dataset contained 342 experimentally determined GPCR structures possessing orthosterically bound non-peptide ligands that were publicly available from the Protein Data Bank [23] as of 31 December 2021 (Appendix A [37,38,39,40,41,42,43,44,45,46,47,48,49,50,51,52,53,54,55,56,57,58,59,60,61,62,63,64,65,66,67,68,69,70,71,72,73,74,75,76,77,78,79,80,81,82,83,84,85,86,87,88,89,90,91,92,93,94,95,96,97,98,99,100,101,103,104,106,111,113,115,116,121,122,123,124,125,126,127,128,129,131,132,133,134,135,136,140,142,145,146,147,148,149,150,151,152,153,158,159,161,162,163,164,166,167,169,170,171,172,173,174,175,176,177,178,179,180,181,182,183,184,185,186,191,192,193,194,196,197,198,200,203,204,205,206,207,208,209,210,211,212,213,214,215,216,217,218,219,220,221,222,223,224,225,226,227,228,229,230,231,232,233,234,235,236,237,238,239,240,241,242,243,244,245,246,247,248,249,250,251,252,253,254,255,256]). Although experimentally determined ligand–receptor complexes are unlikely to be available for GPCR targets with few known ligands, the inclusion of these complexes in the dataset allowed for the classifier to be trained on ligand interactions observed in vitro. Furthermore, the inclusion of experimentally determined GPCR complexes allowed for the classifier’s application to cases where ligand discovery efforts were being made for a target that already possesses resolved structures.

Given that GPCR ligand identification studies often dock potential screening candidates into experimentally determined structures, we aimed to ensure that our classifier was applicable to these cases. Thus, three additional structural categories of ligand–receptor complexes involving experimentally determined structures were included in the internal dataset. To generate the second structural category in the internal dataset, orthosterically bound ligands for each of the 342 experimentally determined structures retrieved from the Protein Data Bank were self-docked into the protein structure they were extracted from, resulting in a structural category comprised of 342 complexes with non-peptide ligands docked into experimentally determined structures. To generate the third structural category in the internal dataset, cross-docking was performed with the orthosterically bound ligands retrieved from the Protein Data Bank for ligands whose target possessed an experimentally determined structure other than the structure the ligand was extracted from. Ultimately, the addition of this structural category to the internal dataset resulted in the inclusion of 326 additional ligand–receptor complexes representing active ligands cross-docked into protein structures that were determined as being in complex with a different ligand.

In addition to using experimentally determined and docked binding modes of known active ligands to develop our random forest classifier, ligand–receptor complexes containing known inactive ligands docked into experimentally determined structures were used to construct the fourth structural category in the internal dataset. To construct this portion of the dataset, the Database of Useful Decoys: Enhanced (DUD-E) [25] was used to obtain inactive ligands and experimentally determined receptor structures for five GPCR targets: adenosine receptor 2 (AA2AR), adrenoceptor beta 1 (ADRB1), adrenoceptor beta 2 (ADRB2), C-X-C chemokine receptor 4 (CXCR4), and dopamine receptor D3 (DRD3). The 285 inactive ligands retrieved from DUD-E (Appendix A) were docked into experimentally determined structures retrieved from DUD-E on a per target basis. The number of inactive ligands docked into each target can be found in Appendix A. Inactive ligand retrieval followed by docking into experimentally determined protein structures resulted in 104 docked complexes for AA2AR, 52 docked complexes for ADRB1, 108 docked complexes for ADRB2, 16 docked complexes for CXCR4, and 5 docked complexes for DRD3.

In addition to docking these known inactive ligands into experimentally determined structures of DUD-E GPCR targets, they were also docked into homology models of DUD-E GPCR targets (the construction of which is discussed in Section 2.2) to generate the fifth structural category in the internal dataset. To maintain a reasonable balance of ligand–receptor complex types (Figure 2) in our internal dataset, one-third of the set of inactive ligands was chosen for docking into homology models using MOE’s Diverse Subset tool. To ensure that the diverse subsets of inactive ligands were reasonably sized per target, the 25 most structurally diverse inactive ligands were selected for docking into targets with >50 inactive ligands (AA2AR, ADRB1, and ADRB2, Appendix A). A diverse subset of inactive ligands was also selected for the 16 inactive CXCR4 ligands, resulting in a subset of 15 inactive ligands for docking into homology models. Since only five inactive ligands were retrieved from DUD-E for DRD3, they were all docked into DRD3 homology models. Inactive ligand docking with each DUD-E target’s set of 3 homology models resulted in 75 docked complexes each for AA2AR, ADRB1 and ADRB2, 45 docked complexes for CXCR4, and 15 docked complexes for DRD3. Examples of top-scoring docking poses resulting from docking AA2AR antagonist compound 10 (Appendix A, Ligand Number 16) and AA2AR inactive 2-amino-6-((2-(2,4-dimethylphenyl)-2-oxoethyl)thio)-4-(thiophen-2-yl)pyridine-3,5-dicarbonitrile (Appendix A, Ligand Number 22) into the AA2AR best-case active template homology model can be found in Appendix A. Although these molecules are differentially characterized in terms of the effect they induce for AA2AR, visual inspection of their binding site placements (Appendix A), MOE-generated ligand interaction diagrams (Appendix A), and structural similarity (as demonstrated by a Tanimoto coefficient of 0.78) may lead one to believe that both molecules possess similar functions and would, thus, be reasonable for prioritization in a virtual screening workflow. However, differences in the actual functions of these molecules justify the development of a machine learning classifier that is trained to recognize differences in specific ligand–residue interactions between functionally different ligands.

To ensure that the classifier was trained on examples of known actives docked into homology models, diverse subsets of known active ligands for each DUD-E GPCR target were docked into each target’s set of homology models to generate the sixth structural category in the internal dataset (Figure 2). First, known active ligands for each of the DUD-E GPCR targets were retrieved from the IUPHAR Guide to Pharmacology (Appendix A) [257]. One third of the set of active ligands retrieved from IUPHAR Guide to Pharmacology was chosen for docking into homology models with the MOE Diverse Subset tool to generate a relatively balanced number of complexes for category 6 (Figure 2). To maintain a reasonable balance of per target representation as well as ligand function in this structural category of the dataset, structurally diverse sets of 9 agonists and 9 antagonists were selected for the 4 targets (AA2AR, ADRB1, ADRB2, DRD3) with both agonists and antagonists listed on the IUPHAR Guide to Pharmacology. Since only 8 antagonists were retrieved for CXCR4, the diverse subset tool was not used when considering which CXCR4 ligands to retain in the subset of active ligands for homology model docking. Active ligand docking with each DUD-E target’s set of 3 homology models resulted in 54 docked complexes per target for AA2AR, ADRB1, ADRB2, and DRD3. For CXCR4, only 24 docked complexes were generated for this portion of the dataset.

### 2.4. External Dataset Preparation

As a further means of assessing the performance of our random forest classifier, an external dataset was constructed. For this dataset, GPR31 and TAAR2 were selected as targets for homology modeling (discussed in Section 2.2) and docking since they are GPCR that lack experimentally determined structures and possess ligands with known function that could be predicted with the random forest classifier. Furthermore, these targets are relatively dissimilar to targets in the internal dataset. For example, global sequence similarities between each external dataset target and the DUD-E GPCR targets represented in the internal dataset (AA2AR, ADRB1, ADRB2, CXCR4, DRD3) ranged from 2.2–12.8% for GPR31 and 3.6–33.4% for TAAR2 (Appendix A).

To construct the external dataset, we first obtained known active and inactive ligands for GPR31 and TAAR2. To obtain a set of GPR31 ligands, we referred to Guo et al.’s 2011 study [27] that screened multiple eicosanoids against cells transfected with GPR31. Based on the screening results presented in this study, we selected three compounds to consider as GPR31 agonists (12(S)-HETE, 5(S)-HETE, and 15(S)-HETE, Appendix A) and one ligand to consider as inactive for GPR31 (12(R)-HETE, Appendix A). For TAAR2, its two endogenous agonists listed in The Concise Guide to Pharmacology 2021/22 [258] (beta phenylethamine and tryptamine, Appendix A) were retrieved and labeled as agonists.

Ligands were then docked into homology models of mixed origin: one was constructed with our in-house GPCR modeling protocol [29,30,31], two were retrieved from GPCRdb [26], and one was retrieved from the AlphaFold Protein Structure Database [33]. In contrast to ligand docking runs used to construct the internal dataset, the use of multiple homology models for external set docking allowed for the generation of a larger external dataset that reflected docked poses of active and inactive ligands in the context of structurally variable receptor models. Altogether, external set ligand docking resulted in the generation of 80 docked complexes for GPR31 and 40 docked complexes for TAAR2. The top-scoring docked poses of GPR31 agonist 12(S)-HETE in complex with the various types of GPR31 modeled structures utilized in the external dataset can be found in Appendix A. Although each docked pose represents the same molecule that has been shown to activate GPR31, one may be inclined to believe that the types of ligand function represented by each docked pose may be different due to differences in binding pocket placement and ligand–receptor contacts. Thus, these exemplary docked poses demonstrate the need to train a classifier capable of identifying active GPCR ligands based on docked or experimentally determined binding modes.

### 2.5. Feature Extraction

Once ligand docking was complete, our internal and external datasets consisted of 1820 and 120 class A GPCR ligand–receptor complexes, respectively. Given that the goal of this work was to develop a method of separating actives from inactives during a virtual screening workflow based on experimentally determined or docked binding modes, our next step was to extract a ligand interaction fingerprint for each ligand–receptor complex. Given that (A) active and inactive state GPCR structures differ in terms of intra-residue contacts [21] and (B) differences in ligand–residue contacts have been observed when comparing GPCR structures bound to ligands of varying functional specificity [259], interaction profiles for each ligand–receptor complex were constructed on a per residue basis. Although the fingerprints used for docking site selection in Section 2.1 could include only interaction sites, machine learning classifiers can consider much richer interaction profiles. The interaction profiles used in this work include interaction energies and interaction types at each site represented in the fingerprint. Non-transmembrane residues (e.g., residues located in extra/intracellular loop regions) in each ligand–receptor complex were not considered when extracting interaction information for a given ligand–receptor complex.

For each ligand–receptor complex in our initial and external datasets with BW numbered residue positions, an interaction profile denoting the types and energetics of ligand–receptor interactions was constructed. Interaction types calculated by the MOE software [35] are categorized as Hbond: hydrogen bonding, Metal: metal–complexed ionic interactions, Ionic: ionic interactions not involving a metal, Arene: pi–pi or cation–pi interactions, or Distance: van der Waals interactions. A single amino acid residue in the receptor can interact with multiple atoms in a ligand, and, thus, may have multiple ligand interaction energies and interaction types. For each BW residue position, feature extraction consisted of obtaining the numeric energetic sum of all interactions occurring at the residue, the categorical type and numeric energy of the most energetically favorable interaction occurring at the residue, and the categorical type and numeric energy of the second most energetically favorable interaction occurring at the residue. Altogether, the following unique interaction types were represented in each dataset: hydrogen bonding, distance-based, arene, ionic, and covalent.

While the BW numbering scheme allowed for a method of indexing transmembrane residue positions, many ligand–receptor complexes in our datasets lacked residues at positions indexed by the numbering scheme due to discrepancies in GPCR sequence length. For example, the first residue position in our dataset for transmembrane domain 1 was indexed as position 1.21, which is 29 residues prior to position 1.50 (representing the most conserved residue in transmembrane domain 1 across class A GPCR). While some GPCRs, such as succinate receptor 1 (PDB identification (PDBID): 6RNK [185]), possess a residue at position 1.21, other GPCRs with fewer residues in transmembrane helix 1 (such as sphingosine 1-phosphate receptor 3, PDBID: 7EW2 [254]) do not possess a residue aligned at position 1.21 (Figure 3). As such, a scheme was constructed to handle cases where a given ligand–receptor complex lacks a residue aligned at a numbered position in our datasets. In addition, this scheme also differentiates the lack of a residue at a BW numbered residue position from cases where a residue is present but makes no interactions with a ligand. If a ligand–receptor complex did not possess a residue at an indexed residue position, its numerical energies and categorical interaction types were assigned as NaN and ‘NA’, respectively. In contrast, if a ligand–receptor complex possessed a residue at an indexed residue position but that did not interact with a ligand, its numerical energies and categorical interaction types were assigned as 0 and ‘None’, respectively. If a residue only possessed a single interaction type, its second interaction type was assigned as ‘None’.

After feature extraction, interaction energies and types were compared between complexes representing actives (those possessing ligands whose function was labeled as agonists, antagonists, or inverse agonists) and inactives in the internal dataset. To avoid including residues that infrequently make interactions in this analysis, only residue positions that possessed interactions in ≥10 (≥0.5%) of the ligand–receptor complexes in our internal dataset (herein referred to as “interacting residue positions”) were selected for further analysis. For each interacting residue position, a mean of all interaction energy sums across all 1820 ligand–receptor complexes was calculated and plotted on a per transmembrane domain basis for complexes with ligands considered as actives or inactives (Appendix A). In addition, the percentage of ligand–receptor complexes possessing an interaction at each interacting residue position was calculated and plotted on a per transmembrane domain basis for complexes with ligands considered binders or non-binders (Appendix A). Interaction percentages were also calculated for each interacting residue position per interaction type (hydrogen bonding, distance-based, arene, ionic, and covalent (Appendix A). While there are some noticeable differences per interacting residue position in interaction energy sum means (such as residue position 3.32 exhibiting a more negative interaction energy sum for complexes containing actives, Appendix A), interaction percentages (such as residue positions 2.54 and 2.56 interacting in complexes containing actives but not interacting in complexes containing inactives, Appendix A), and per type interaction percentages (such as arene interactions occurring more frequently at residue position 3.28 for complexes of inactives, Appendix A) when active and inactive complexes are compared, there was no single variable/set of variables that allowed for surface-level separation of active/inactive ligands. Thus, machine learning was explored to differentiate active complexes from inactive complexes.

Prior to developing the machine learning classifier, we desired to remove interaction energies and types for BW indexed residue positions that either infrequently appeared in GPCR structures in the internal dataset (due to the helix length-dependent nature of the BW numbering scheme) or infrequently interacted with a ligand. As such, a residue position was only retained for feature set consideration if more than 10 ligand–receptor complexes in the internal dataset possessed non-NA (since an interaction energy sum value of ‘NA’ was assigned when a receptor lacked a residue at a BW indexed position) or nonzero (since an interaction energy sum value of zero was assigned when a receptor lacked a residue at a BW indexed position) interaction energy sums at that position. Altogether, removal of infrequently appearing/interacting residue positions resulted in a set of 63 BW indexed residue positions whose interaction energies and types were used as predictors for the random forest classifier (Appendix A).

### 2.6. Ligand Function Prediction

#### 2.6.1. Four Functional Class Predictions

We initially developed a random forest classifier to predict one of four ligand functions based on interaction profiles. Ligand functions of agonist, inverse agonist, antagonist, or inactive were based on the activity types listed on GPCRdb [26]. Our internal dataset was split into training and testing datasets using a 75%/25% split, resulting in training and testing sets containing 1365 and 455 samples, respectively. The distribution of ligand–receptor complex structural categories in our training and testing datasets was like that in the internal dataset (Figure 4), ensuring that classifier performance could be assessed for a variety of GPCR ligand–receptor complex types. After the splitting of the internal dataset, the resulting training and testing datasets were each subjected to preprocessing in the form of the imputation of missing values and standardization.

After preprocessing of the training and testing datasets, a random forest classifier was developed using 10-fold cross-validation to avoid overfitting. Once the random forest classifier was trained, ligand functions for ligand–receptor complexes in the testing set were predicted. A confusion matrix containing ligand function prediction results for the test set as well as training/testing set classifier performance metrics can be found in Table 5. Cross-validation resulted in a mean score of 0.80, indicating that the classifier did not overfit since ligand functions were accurately predicted for examples excluded during classifier fitting. In terms of the classifier’s performance with respect to the testing set, ligand function prediction for testing set ligand–receptor complexes resulted in an accuracy value of 0.76, indicating that the random forest classifier accurately predicted ligand functions for 76% of complexes in the testing dataset. Furthermore, testing set classification resulted in a precision value of 0.76, indicating that 76% of samples predicted to be in each class of ligand function were actual positives for each class of ligand function. Testing set classification also resulted in a recall value of 0.76, indicating that 76% of actual positives for each class of ligand function were predicted correctly.

The random forest classifier was also used to predict ligand functions for the external dataset containing 120 ligand–receptor complexes that resulted from docking 6 GPR31/TAAR2 ligands of varying function into homology models from various sources. Classification performance was again assessed using accuracy, precision, and recall metrics. In contrast to the encouraging results observed with the testing set, initial classification performance with the external dataset was poor (Table 6). Use of the random forest classifier to predict ligand function for the 120 ligand–receptor complexes in the external dataset led to observed accuracy, precision, and recall metric values of 0.03, indicating that the classifier did not generalize well to the external dataset.

In contrast to the internal dataset that was used to train and test the random forest classifier, each ligand–structure pairing in the external dataset was represented by a set of five retained docked poses (rather than a single retained pose per ligand–structure pairing as observed in the internal dataset). Consequently, we recognized a potential challenge in a situation where this classifier would be applied to docking results for a target GPCR during a computational ligand identification study: if five docked poses are generated and classified per GPCR model–ligand pairing, how does one handle a case where separate docked poses of the same ligand are classified differently (i.e., pose 1 is classified as an agonist and pose 2 is classified as inactive) within the same modeled structure? Thus, an additional set of predictions was made for the external dataset using majority rule voting to assess classifier performance on a per ligand (rather than per docked pose) basis. For each GPCR model–ligand pairing in the external dataset, its set of five predictions (representing one prediction of ligand function per docked pose) was used to determine a “majority” prediction based on the ligand function that was more frequently predicted across its five docked poses. For example, a ligand with three docked poses predicted as agonist and two docked poses predicted as inactive would be considered an agonist using this scheme. After calculating majority predictions for each GPCR model–ligand pairing in the external dataset, classification performance was again assessed using the accuracy, precision, and recall metrics (Table 6). Despite this initial reconfiguration of predictions for the external dataset, use of the random forest classifier to predict ligand function for each of the 24 GPCR model–ligand pairings in the external dataset using majority rule voting led to observed accuracy, precision, and recall metric values of 0.04, again indicating that the classifier did not generalize well to the external dataset.

#### 2.6.2. Two Functional Class Predictions

Although initial classification of the external dataset led to discouraging results, we found it promising that active ligands (agonists, antagonists, and inverse agonists) were infrequently predicted to be inactive when classifying the testing and external datasets. For example, 19 of the 20 external set **GPCR model–ligand** pairings possessing a docked agonist were predicted as antagonists with majority rule predictions (Table 6). Even though the predicted functions for these ligands were incorrect, they were still predicted to interact with the target receptor. Given this, we hypothesized that reconfiguring ligand function predictions from four classes (agonist, antagonist, inverse agonist, and inactive) to two classes (where agonists, antagonists, inverse agonists are merged into the active class with no change in the inactive class) after function prediction may result in more accurate ligand function prediction. Although reducing the number of ligand function classes from four to two results in a less detailed prediction of how a ligand may or may not induce a change in receptor activation state, we found this reconfiguration of predictions to be justified, since most early efforts to identify GPCR ligands are considered successful upon identification of ligands that bind the target receptor, regardless of which type of change in receptor function is induced. Furthermore, this reconfiguration allowed for the calculation of a hit rate metric (Equation (1)) that denoted the percentage of predicted actives that were actual actives. We found this metric to be quite important in the context of GPCR ligand identification, since the metric allows for an approximation of the proportion of ligands likely to be actives in the set of ligands selected for experimental screening based on predicted activity. Equation (1) is as follows:(1)$$\mathrm{Hit}\;\mathrm{rate}\;(\%)=100\;\times \;\frac{\mathrm{Actual}\;\mathrm{Actives}\;\mathrm{within}\;\mathrm{Set}\;\mathrm{of}\;\mathrm{Predicted}\;\mathrm{Actives}}{\mathrm{Predicted}\;\mathrm{Actives}}$$

A reassessment of classification performance for merged active testing set predictions can be found in Table 7. When test set classification results were reconfigured into two classes, accuracy increased from 0.76 (Table 5) to 0.85 (Table 7). Values for the precision and recall also increased from 0.78 to 0.85 and 0.76 to 0.85, respectively. Additionally, a hit rate of 95.4% was observed. Altogether, the increases in classification performance metrics values as well as the remarkably high hit rate resulting from merged active predictions indicated that ligands predicted as actives with our classifier are likely to be actual actives.

Given that both experimentally determined and modeled structures were used to construct the testing dataset, classification performance metrics and hit rates were also calculated for each structure type. We found this analysis to be particularly important in the context of ligand identification efforts for understudied GPCRs, since these efforts typically involve the use of homology models due to the lack of a published structure. The experimentally determined testing set was comprised of the following structure types from our internal dataset (Figure 2): experimentally determined structures of binders retrieved from the PDB, binders self-docked into experimentally determined structures, binders cross-docked into experimentally determined structures, and non-binders docked into experimentally determined structures of DUD-E GPCR targets. After the prediction of ligand function and subsequent merging of active predictions for ligand–receptor complexes involving experimentally determined structures, accuracy, precision, and recall values of 0.96 were observed (Table 8), indicating that merging active predictions after initial classification led to accurate predictions of ligand function for ligand–receptor complexes concerning experimentally determined structures. Furthermore, the observed hit rate of 97.6% indicated that most ligands that were predicted as actives after prediction reconfiguration were actual actives. When comparing classifier performance metrics between the set of initial predictions and the set of merged active predictions for ligand–receptor complexes concerning experimentally determined structures, it is evident that merging actives after making initial predictions with the classifier led to more accurate predictions due to greater observed accuracy, precision, and recall values (Table 8).

Classification performance metrics and hit rates were also calculated for ligand–receptor complexes in the testing set that involved modeled structures (Table 9). The modeled structure testing set was comprised of the following structure types from our internal dataset (Figure 2): inactives docked into homology models of DUD-E GPCR targets and actives docked into homology models of DUD-E GPCR targets. After the prediction of ligand function and subsequent merging of active predictions for ligand–receptor complexes involving homology models, accuracy, precision, and recall values of 0.60 were observed, indicating that classifier performance was worse for ligand–receptor complexes involving homology models when compared to the classifier performance for ligand–receptor complexes involving experimentally determined structures (accuracy, precision, and recall = 0.96 for the experimentally determined testing set, Table 8). Additionally, merged active predictions resulted in a hit rate of 80.6%, which is worse than the hit rate observed with merged active predictions for the experimentally determined testing set (hit rate = 97.6%, Table 8) but ultimately indicates that about four of the five ligands in complex with homology models that were predicted to be active are true actives. Although classification performance with merged active predictions was worse for ligand–receptor complexes involving homology models, we found this result to be acceptable since (A) the homology model testing set only contained docked poses of known active and inactive ligands (in contrast to the experimentally determined testing set which contained GPCR ligand binding modes retrieved from the PDB) and (B) discrepancies in homology model quality from target to target inherently induces experimental error. To further justify the use of merged active predictions, classifier performance metrics were again compared between the set of initial predictions and the set of merged active predictions for ligand–receptor complexes involving modeled structures. Accuracy, precision, and recall values resulting from the merged active predictions (0.60 for each metric, Table 9) were all greater than those resulting from initial predictions (0.52 for each metric, Table 9) for the homology model testing set, indicating that merging actives after initial predictions were made led to improved prediction of ligand function.

Lastly, a set of merged active predictions was made for the 24 **GPCR model–ligand** pairings (six GPR31/TAAR2 ligands, four structures per target) in the external dataset using the majority rule classification scheme previously detailed in this section. After predictions were made, classifier performance was assessed with the hit rate, accuracy, precision, and recall metrics (Table 10). In contrast to the poor classifier performance observed when ligand functions were initially predicted for ligand–receptor complexes in the external dataset with majority rule voting (Table 6), reconfiguring the ligand function predictions from four classes to two classes led to much better performance with the external dataset. Accuracy, precision, and recall values were all greater for the set of merged active predictions (0.79 for each metric, Table 10) than the set of initial predictions (0.04 for each metric, Table 6), which further supported the reconfiguration of initial classifier predictions into two classes. Furthermore, the observed binder hit rate of 82.6% (Table 10) indicated that the classifier identified a large proportion of true actives in the set of external dataset ligands predicted to be actives. Of some concern is the failure of the classification scheme to correctly classify the inactive ligand (regardless of homology model it was docked into). In a typical virtual screening workflow, the potential screening candidates are likely to include more inactive than active structures. If these are not correctly predicted as inactive, lower hit rates might be observed. However, with only one inactive compound in the dataset due to limited published screening data at these targets, the ability of the classifier to identify inactives was inadequately tested with this external dataset. Given that 60 out of 66 inactives were correctly predicted in the internal testing set (Table 8), it seems likely that the classifier will prove useful to enrich screening sets in VS workflows. Altogether, we suggest the use of majority rule predictions if multiple docked poses are to be classified per ligand when applying this classifier to a GPCR ligand identification workflow.

Our use of modeled structures from multiple sources (in-house [29,30,31], AlphaFold [32,33], and GPCRdb [26]) when docking known active and inactive ligands for the two external set targets (GPR31, TAAR2) also enabled a comparison of external set classification performance between model types. For each of the six GPR31/TAAR2 ligands docked into the in-house, AlphaFold, or GPCRdb active template, or the GPCRdb inactive template homology models and classified with a combination of merged active and majority rule prediction methodologies, a confusion matrix was generated to assess external set classification performance using hit rates in the context of each modeled structure source (Table 11). Hit rates were comparable between the in-house, AlphaFold, GPCRdb active template and the GPCRdb inactive template homology models (83.3%, 80.0%, 83.3%, and 80.0%, respectively, Table 11), indicating that the classifier was able to correctly identify active ligands docked into a variety of modeled structures for the targets in the external dataset.

## 3. Materials and Methods

In this work, we aimed to develop a classifier that can predict whether any given ligand is likely to be active through interaction at the orthosteric binding site of a given GPCR target based on its experimentally determined or docked GPCR complex structure, and what type of function that ligand would produce if active. In this context, active molecules induce one of three changes in receptor function. Active molecules that stimulate GPCR signaling are agonists, those that block agonist-stimulated GPCR signaling without impact on basal GPCR activity are antagonists, and those that reduce basal GPCR signaling are inverse agonists. Molecules that do not induce any of these changes in receptor function are considered inactive. An ideal classifier would be able to use complexes generated by classical docking to predict the activity and function of ligands not yet synthesized or tested for activity. Thus, ligand interaction fingerprints were developed and assessed as docking site selection tools to determine whether such fingerprints improve pose prediction prior to the development of machine learning classifiers for ligand activity prediction.

### 3.1. G Protein-Coupled Receptor (GPCR)–Ligand Interaction Fingerprints

Interactions between GPCR structures and bound ligands were calculated for each PDB [23] entry shown in Appendix A using output generated by the RING Server [36]. RING Server output provided ligand interaction sites and distances. Distance-based interaction scores were calculated [30] using distance data produced by the RING Server. Distances greater than 4.63 Å were assigned interaction scores of 0. Distances less than 3.23 Å were assigned interaction scores of 1. Interaction scores for distances between these ranges were linearly scaled between 0 and 1. Interaction sites were numbered using the BW system to make three-dimensionally relevant comparisons across different members of the GPCR family. Weighted interaction frequencies were calculated for each receptor site to ensure that each of the 60 GPCR family members represented by differing numbers of ligand–receptor complex structures had equal weight in the fingerprint. These weighted interaction frequencies were calculated using Equation (2) in which n is the number of receptor–ligand complexes contributing to the fingerprint, *I_i_* is 1 if the site does and 0 if the site does not interact with the ligand in complex *i*, and *m* is the number of complexes containing the same GPCR family member as receptor–ligand complex *i*.
(2)$$\mathrm{Weighted}\;\mathrm{interaction}\;\mathrm{frequency}=\frac{\sum_{i=1}^{n}\frac{I_{i}}{m}}{n}$$

Weighted interaction frequencies can range from 0 for sites that do not interact with ligand in any of the receptor–ligand complexes to 1 for sites that interact with ligands in all receptor–ligand complexes. Different fingerprints were derived from interaction frequencies for the entire set of structures in Appendix A without applying an interaction score cutoff (fingerprint A) and with the application of an interaction score cutoff of 0.5 to emphasize stronger interactions (fingerprint B). Different fingerprints were derived from each subset of structures in Appendix A with common activation states (active fingerprint, inactive fingerprint, intermediate fingerprint) and an interaction score cutoff of 0.5. Interaction sites were included in the various fingerprints based on a threshold weighted interaction frequency at that site. Thresholds were selected to include 10–15 interaction sites per fingerprint and ranged from 0.35 for the intermediate fingerprint to 0.6 for the global interaction fingerprint with no interaction score cutoff.

### 3.2. Ligand Docking to Assess Ligand Interaction Fingerprints

The structures used as input for docking experiments were obtained from the PDB [23]. The most complete GPCR chain and associated orthosteric ligand were retained and any other entities were deleted. The structure preparation operation in the MOE (version 2019) software was utilized to add hydrogens, missing sidechains, cap amino acids adjacent to uncharacterized loops, and atomic charges. Ligand functional groups with pKa values near 7 were docked in both acid and conjugate base forms. Reference docking calculations utilized the SiteFinder function in the MOE software [35] to select amino acid residues surrounding the orthosteric site as the docking site. Fingerprint docking calculations utilized amino acid residues in the identified fingerprint as the docking site. Each docking calculation generated 1000 initial poses, of which the top 400 were refined using the induced fit setting which allowed sidechain flexibility in the vicinity of the ligand and recomputed complex energies using the GBVI scoring method. The 5 lowest energy poses (top 5) were retained after final scoring for assessment of fingerprints (scoring accuracy). All 400 refined poses were retained to determine sampling accuracy. The docking calculations used to test the suitability of fingerprints as docking site selection tools utilized structures from two different PDB entries containing the same GPCR. The protein structure was extracted from one PDB entry and the ligand structure was extracted from the second PDB entry. The two PDB entries utilized serve as cross-docking structure pairs. The cross-docking structure pairs utilized to test each fingerprint are identified in Appendix A. Docking accuracy was determined by superposition of the alpha carbon atoms in the protein chain of the docked complex onto the alpha carbon atom positions of the reference complex (PDB entry used as the ligand structure source), followed by the calculation of the root-mean-square deviation (RMSD) of ligand heavy atom positions between the docked position and the reference position. The lowest RMSD obtained from all of the ways to compare symmetrical heavy atoms (such as all three methyl carbon atoms of a tertiary-butyl group) is reported.

### 3.3. Overview of Datasets to Train and Test Machine Learning Classifiers

In total, 1820 class A GPCR ligand–receptor complexes were used as an internal dataset from which to train and test machine learning classifiers for ligand activity prediction. Ligand–receptor complexes in the internal dataset were grouped into 6 categories (Figure 1) based on structure type (experimentally determined or modeled) as well as ligand activity (active or inactive):Experimentally determined structures of actives;Actives self-docked into experimentally determined structures;Actives cross-docked into experimentally determined structures;Inactives docked into experimentally determined structures of GPCR targets listed on DUD-E [25];Inactives docked into homology models of DUD-E GPCR targets;Actives docked into homology models of DUD-E GPCR targets.

The following sections detail the construction of the internal dataset as well as the development of machine learning classifiers used for ligand function prediction.

### 3.4. Acquisition of Ligands and Experimentally Determined Structures

All 342 class A GPCR structures possessing orthosterically bound non-peptide ligands that were publicly available in the PDB [23] as of 31 December 2021 were downloaded (Appendix A) as a source of active ligands and structures for docking. These 342 experimentally determined structures comprised the first category of ligand–receptor complexes in the internal dataset (experimentally determined structures of actives) and were used to generate the second category (actives self-docked into experimentally determined structures).

Next, structures for the third category, actives cross-docked into experimentally determined structures (wherein a ligand is docked into an alternate structure of the target it was extracted from), were obtained (Appendix A). For each of the active ligands used for docking into experimentally determined structures, the UniProt name of the GPCR that the ligand was extracted from was used to obtain an initial list of additional structures of the same GPCR from GPCRdb [26]. From this list of additional structures, the structure that possessed the lowest resolution and matched the activation state of the complex containing the ligand was selected for cross-docking. If a structure meeting the latter criterion was not available, then the lowest resolution structure was selected for docking. If an additional experimentally determined structure was not available for a target, then cross-docking was not performed.

A set of inactive ligands for docking into experimentally determined structures of DUD-E GPCR targets was needed to generate the fourth and fifth sets of complex structures in the internal dataset. Known inactive ligands for 5 GPCR targets (adenosine receptor 2 (AA2AR), adrenoceptor beta 1 (ADRB1), adrenoceptor beta 2 (ADRB2), C-X-C chemokine receptor 4 (CXCR4), and dopamine receptor D3 (DRD3)) were retrieved from DUD-E [25] (Appendix A, 285 compounds). In addition, experimentally determined structures of each of the 5 GPCR targets were retrieved from DUD-E and saved to their own database. After making considerations based on the size of each target’s set of inactive ligands, MOE’s Diverse Subset tool was then used on the set of 285 known inactive ligands to select a subset of 95 structurally diverse inactive ligands for docking into modeled structures of the DUD-E targets. Diverse subset calculations were performed on a per target basis to ensure that each target was fairly represented in the subset of 95 structurally diverse inactive ligands.

To create the set of active ligands used for docking into modeled structures (Appendix A), the IUPHAR/BPS Guide to Pharmacology [257] was used to obtain active agonists and antagonists for each of the 5 GPCR targets with inactives listed on DUD-E based on the following criteria:The agonist/antagonist is active at the human ortholog of the target GPCR;The agonist/antagonist is a non-peptide ligand;The agonist/antagonist is not radiolabeled;The agonist/antagonist is not an allosteric modulator.

Once an initial set of agonists and antagonists was retrieved, the MOE [35] Diverse Subset tool was used to obtain the final set of ligands to be docked into each target. If a target’s respective set of agonists or antagonists possessed > 9 ligands, the Diverse Subset tool was used to obtain the 9 most structurally dissimilar compounds for docking. In total, 80 known active ligands for DUD-E GPCR targets were selected for docking.

To validate the performance of our random forest classifier with an external dataset, G protein-coupled receptor 31 (GPR31) and trace amine-associated receptor 2 (TAAR2) were selected as targets for which to predict ligand function. To obtain GPR31 actives and inactives, 4 ligands screened against GPR31 in a 2011 publication by Guo et al. [27] (3 active agonists and 1 inactive) were retrieved and labeled according to their function within a database. Next, 2 endogenous TAAR2 agonists listed in The Concise Guide to Pharmacology 2021/22 for GPCR [258] were also retrieved and labeled according to their function within a database.

### 3.5. Protein Modeling

To represent cases where GPCR lack an experimentally determined structure in our internal dataset, loop refined homology models were generated for the 5 GPCR targets listed on DUD-E using our previously benchmarked GPCR modeling workflow that retains the crystallized template ligand throughout the homology modeling and extracellular loop 2 (ECL2) modeling processes [29,30,31]. Template structures from which to model each target were selected using the contact-informed neighboring pocket (CoINPocket) score to emphasize similarities at residue positions that frequently make strong interactions in a set of 27 unique class A GPCR experimentally determined structures [30]. For each target, 3 template structures were selected from which to construct 2 “best case” homology models (wherein selected template receptors bind the same endogenous ligand as the target and represent active and inactive receptor states) and 1 “normal case” homology model (wherein a selected template receptor does not bind the same endogenous ligand as the target) (Table 4). Prior to the construction of a target’s best case homology models, active and inactive state structures of the GPCR possessing the highest CoINPocket similarity score to the target were selected as templates for homology model construction. For each target’s normal case homology model, the GPCR possessing the highest CoINPocket similarity score to the target that did not bind the same endogenous ligand as the target was selected as a template structure. After template selection, a set of 11 initial homology models was generated for each target–template pairing using our previously benchmarked GPCR modeling workflow [29,30,31]. This workflow utilizes MOE’s default homology modeling settings, save for scoring models based on effective contact energy and retaining the crystallographic ligand from the template structure as the ‘Environment for Induced Fit’. For each target–template pairing, the homology model with the lowest effective contact energy was selected for de novo extracellular loop 2 (ECL2) modeling.

Prior to ECL2 modeling, the final helical residue of TM4 and first helical residue of TM5 were selected as loop ‘anchor’ residues (Appendix A). Next, fragment libraries (which are a requirement for Rosetta’s “kinematic closure with fragments” (KICF) [260] ECL2 sampling method used in this work) were generated by submitting a FASTA formatted sequence containing the nine residues prior to the first loop anchor, the ECL2 sequence and the nine residues after the second loop anchor to the *Robetta* [261] server. Loop modeling incorporated an atomic disulfide constraint that restricted the distance of sulfur atoms in critical cysteine residues 3.25 of TM3 and 45.50 of ECL2 to 5.1 Å as a means of filtering out models unable to form disulfide bonds. Furthermore, the ligand from the template structure was retained in the homology model binding pocket during loop modeling. For each target, a total of 250 disulfide-constrained ECL2 models were generated. The ECL2-TM3 disulfide bond was formed in the top 10 lowest scoring models followed by geometry optimization of the ECL2 segment in MOE. Each target’s lowest scoring loop refined homology model was then selected for ligand docking.

For each target in our external dataset, a set of protein models containing in-house homology models, GPCRdb [26] homology models, and an AlphaFold [32] model served as docking target structures. First, in-house homology models were constructed for each target with our benchmarked GPCR modeling protocol (Appendix A). Next, homology models for each target were retrieved from the AlphaFold Protein Structure Database [33] and GPCRdb [26]. For GPCRdb homology models [201], both active and inactive state template structure homology models were retrieved for GPR31 and TAAR2. Once retrieved and/or constructed, all homology models for targets in our external validation set were stored in their own respective databases for docking.

### 3.6. Ligand Docking to Generate Complexes Used to Train and Test Machine Learning Classifiers

Prior to ligand docking, each ligand database was prepared at pH 7.4 using the QuickPrep function in MOE [35] to ensure proper protonation and charge at the desired pH and perform energy minimization using the AMBER10:EHT forcefield [262]. All receptor structures for docking were also prepared with the QuickPrep function. For each receptor structure, a prospective binding site was then defined with MOE’s SiteFinder function [35]. This function organizes potential binding sites by the volume of alpha spheres within a potential binding pocket, based on the Alpha Shapes methodology of Edelsbrunner et al. [34]. Once a structure’s binding site was defined, ligands were docked into each structure using MOE-induced fit docking, which initially placed and optimized 1000 poses without receptor sidechain flexibility. Next, the top 400 poses (based on the London dG scoring function) were passed on to the refinement stage, which incorporates flexible protein sidechains and uses the generalized Born volume integral/weighted surface area (GBVI/WSA) scoring function [263]. For the internal dataset, only the best scoring pose from the refinement stage was retained for feature extraction.

The ligands docked into structures used to generate ligand–receptor complexes in the internal dataset can be found in Appendix A. For the set of active ligands to be docked into experimentally determined structures (Appendix A), each ligand was redocked into the experimentally determined structure it was extracted from, resulting in a set of 342 docked poses in ligand–receptor complex category 2. Each of these actives was also cross-docked (Appendix A) to produce its corresponding cross-docking structure (if applicable), resulting in a set of 326 docked poses in ligand–receptor complex category 3.

In addition, each of the 285 known inactive ligands for the 5 DUD-E GPCR targets were docked into experimentally determined structures retrieved from DUD-E on a per target basis, resulting in a set of 285 docked complexes in ligand–receptor complex category 4. For the structurally diverse set of 95 known inactive ligands for the 5 DUD-E GPCR targets (denoted by asterisks in Appendix A), each ligand was docked into its target’s best- and normal-case homology models, resulting in the set of 285 docked complexes comprising ligand–receptor complex category 5. For the structurally diverse set of 80 known active ligands for the 5 DUD-E GPCR targets (Appendix A), each ligand was docked into its target’s best- and normal-case homology models, resulting in the set of 240 docked complexes comprising ligand–receptor complex category 6.

The ligands docked into TAAR2 and GPR31 modeled structures to produce the external dataset are shown in Appendix A. For the set of 4 ligands to be docked into GPR31, each ligand was separately docked into each of the 4 constructed or retrieved GPR31 models. For the set of 2 active ligands to be docked into TAAR2, each ligand was separately docked into each of the 4 constructed or retrieved TAAR2 models. In contrast to docking performed to construct the internal dataset, docking runs used to generate the external dataset retained the 5 best poses after the refinement stage (rather than a single pose). Altogether, our external validation dataset contained 120 docked complexes for GPR31 and TAAR2.

### 3.7. Feature Extraction

Prior to feature extraction, GPCR residue positions for each ligand–receptor complex in our dataset were first indexed using the BW numbering scheme.

Once residue positions were indexed for transmembrane residues in each ligand–receptor complex, feature extraction was performed to obtain categorical interaction types and numerical interaction energies at each residue position. For each ligand–receptor complex, the following information was extracted for each indexed residue position:The energetic sum of all interactions occurring at the residue (which may include interactions of different atoms in the amino acid residue with different atoms in the ligand;The categorical type of the most energetically favorable interaction occurring at the residue (Hbond: hydrogen bonding, Metal: metal–complexed ionic interactions, Ionic: ionic interactions not involving a metal, Arene: pi–pi or cation–pi interactions, or Distance: van der Waals interactions);The numerical energy of the most energetically favorable interaction occurring at the residue;The categorical type of the second most energetically favorable interaction occurring at the residue;The numerical energy of the second most energetically favorable interaction occurring at the residue.

Although all transmembrane residues in a given ligand–receptor complex can be indexed with the BW numbering scheme, a majority of transmembrane residues do not interact with a ligand. If an indexed residue did not possess interactions with a ligand, its energies were denoted as zero and its interaction types were denoted as ‘None’. If a ligand–receptor complex lacked a residue at a BW numbered index position (which results due to discrepancies in transmembrane domain lengths between GPCR), its interaction energies and types were denoted as ‘NA’. Once the internal dataset denoting the types and energies of interactions at each BW indexed residue position was created, each ligand–receptor complex was assigned to one of four target classes (agonist, antagonist, inverse agonist, inactive) based the activity type of the ligand in the complex. Feature extraction to obtain interaction energies and types was also performed for the external dataset prior to its classification.

### 3.8. Ligand Activity Prediction

In this work, a classifier capable of predicting ligand activity based on per residue interaction energies and types extracted from a given ligand–receptor complex was developed. This classifier was written in Python 3.9.7 and developed using the *RandomForestClassifier* algorithm contained within the Scikit-learn version 0.24.2 machine learning library [264]. This classifier has been made freely available at https://github.com/gszwabowski/GPCR_DB_project, accessed on 23 May 2024.

#### 3.8.1. Data Preprocessing

To select a set of residue positions whose interaction energies and types would be used as random forest classifier features, the number of internal dataset ligand–receptor complexes possessing interactions with a ligand at each residue position was calculated. The final list of residue positions whose interaction energies and types were used to predict ligand function can be found in Appendix A. In addition to using interaction energies and types for residue positions as predictors for the random forest classifier, an additional predictor denoting whether a ligand–receptor complex contains a modeled protein structure was added to the dataset. Ligand–receptor complexes containing modeled protein structures were assigned a value of 1, while ligand–receptor complexes containing experimentally determined protein structures were assigned a value of 0.

After determining a final set of predictors for classifier development via interaction percentages, categorical variables within the internal dataset (interaction types) were ordinally encoded as integers. In addition, target classes representing ligand function were removed from the dataset prior to classification. Next, the internal dataset was split into training and testing subsets using a randomized 75% to 25% train to test split. Prior to classifier development, missing values within numerical columns of the training and testing subsets were imputed with Scikit-learn’s *SimpleImputer* function, which replaced missing values using the mean along each column [264]. The training and testing subsets were then standardized with Scikit-learn’s *StandardScaler* function, which shifted the distribution of all input variables to have a mean of 0 and a standard deviation of 1 [264].

Prior to classification of the external dataset, the predictor denoting whether a ligand–receptor complex contains a modeled protein structure was added to the dataset. Additionally, the external dataset was also subjected to ordinal encoding of categorical variables as well as scaling and imputing with Scikit-learn’s *SimpleImputer* and *StandardScaler* functions, respectively.

#### 3.8.2. Random Forest Classifier Development

The ligand function prediction for each ligand–receptor complex was performed using the *RandomForestClassifier* algorithm contained in the Scikit-learn machine learning library [264]. This algorithm is rooted in the concept of random forests, an ensemble learning approach that constructs many decision trees and outputs predictions based on predicted class votes of the individual decision trees [24]. With this approach, randomly sampled subsets of the training data are chosen from which to grow individual decision trees. Each individual decision tree is then grown by splitting its subset of training data at each node (i.e., “branching” the decision tree) according to a feature sampled from a random subset of training data features. In contrast to traditional implementations of a random forest where each classifier is allowed to vote for a single class, Scikit-learn’s *RandomForestClassifier* algorithm instead combines individual classifiers by averaging their probabilistic predictions for classification [264].

We applied random forest classification modeling to predict the activity of ligands complexed with experimentally determined structures or docked into experimentally determined or modeled structures using interaction energies/types for 63 BW numbered residue positions and a binary indicator denoting whether a complex involves a modeled protein structure as features. Once data preprocessing of the internal dataset was complete, a random forest classifier implementing 10-fold cross-validation was trained on our training subset using parameters tuned with Scikit-learn’s *GridSearchCV* function. The final parameters for the random forest classifier were as follows: *n_estimators* = 500, *class_weight* = ‘balanced_subsample’, *bootstrap* = False, *max_depth* = 30, *max_features* = ‘auto’, *min_samples_leaf* = 1, and *min_samples_split* = 2.

The trained random forest classifier was then used to predict ligand function for the 455 testing subset samples. In our initial set of predictions, ligands were classified as agonists, antagonists, inverse agonists, or inactives. In addition to the set of predictions resulting from initial classification, a modified set of predictions that reconfigured predictions into binary categories of ligand activity was created. In this reconfiguration of our initial predictions (herein referred to as our “merged active” prediction set), any ligand predicted to be an agonist, antagonist, or inverse agonist was relabeled as an active, while any ligand predicted to be inactive was not relabeled. Furthermore, 2 additional subsets of the merged active prediction set were created that individually represented predictions made for ligands complexed with experimentally determined protein structures or predictions made for ligands complexes with modeled protein structures. For each set or subset of predictions, performance of the classifier was assessed with the precision (Equation (3)), accuracy (Equation (4)), and recall (Equation (5)) metrics, which are given by the following:(3)$$\mathrm{Precision}=\frac{\mathrm{TP}}{\left(\mathrm{TP}+\mathrm{FP}\right)}$$
(4)$$\mathrm{Accuracy}=\frac{\mathrm{TP}+\mathrm{TN}}{\left(\mathrm{TP}+\mathrm{FP}+\mathrm{FN}+\mathrm{TN}\right)}$$
(5)$$\mathrm{Recall}=\frac{\mathrm{TP}}{\left(\mathrm{TP}+\mathrm{FN}\right)}$$
where TP, TN, FP, and FN are the numbers of true positives, true negatives, false positives, and false negatives, respectively, for each class of ligand activity. In addition, hit rates were calculated for the merged active prediction sets using Equation (1).

## 4. Conclusions

In this work, a random forest classifier was developed to predict whether a potential orthosteric GPCR ligand is likely to be considered active (agonists, antagonists, or inverse agonists) or inactive (wherein a molecule does not induce any change in GPCR function) based on per residue interaction energies and types extracted from its experimentally determined or docked binding mode. Docked binding modes were determined using classical docking using automated site selection after ligand interaction fingerprints were developed and found to offer modest advantages in sampling accurate poses, but without substantial advantage in the final set of top-ranked poses after scoring. The classifier was first trained to predict 4 functional classes (agonist, antagonist, inverse agonist, inactive) using interaction profiles extracted from 1365 class A GPCR ligand–receptor complexes that represented a diverse variety of structure types typically encountered in GPCR ligand identification studies (Figure 4A), and it was validated using 10-fold cross-validation. Subsequent classification of the 455 class A GPCR ligand–receptor complexes in the testing set (Figure 4B) was assessed using the accuracy, precision, and recall metrics. Initial training and testing of the classifier led to a cross-validation score of 0.80 and accuracy, precision, and recall values of 0.76 (Table 5), indicating that the classifier did not overfit to the testing dataset and was capable of accurately predicting ligand function across a variety of GPCR complex types.

The classifier was also used to predict 4 classes of ligand function for an external dataset containing 120 ligand–receptor complexes that resulted from docking 6 GPR31/TAAR2 ligands of varying function (Appendix A) into in-house, GPCRdb [26], and AlphaFold [32,33] homology models. Initial classification results for the external dataset were poor, as exhibited by accuracy, precision, and recall metric values of 0.03 when predicting the function for each of the 120 external set ligand–receptor complexes (Table 6). In addition, we desired to address the hypothetical case where separate docked poses of the same **GPCR model–ligand** pairing in the external dataset are contradictorily predicted. Thus, predictions were also made for the external dataset using majority rule voting (where a **GPCR model–ligand** pairing was assigned a predicted function based on the function most frequently predicted in its set of five docked poses). However, four class predictions with majority rule voting also resulted in poor accuracy, precision, and recall values of 0.04 (Table 6).

However, the poor performance of the classifier when predicting four functional classes for the external dataset led to our hypothesis that reducing the number of functional classes from four (agonist, antagonist, inverse agonist, inactive) to two (active, inactive) would lead to more accurate predictions of ligand function. Thus, initial predictions for the testing and external datasets were reconfigured with our “merged active” methodology, where agonists, antagonists, and inverse agonists were merged into the active class with no change in the inactive class ligands. For the testing set, merged active reconfiguration of initial predictions resulted in increases in accuracy, precision, and recall metric values (accuracy, precision, and recall = 0.85, Table 7) when compared to initial predictions (accuracy, precision, and recall = 0.76, Table 5), indicating that merging active classes after making initial ligand function predictions led to better classifier performance. Furthermore, reconfiguration of ligand function prediction into a binary classification problem allowed for the calculation of a hit rate metric (Equation (1)) that denoted the percentage of ligands predicted to be actives that were actual actives. A hit rate of 95.4% was observed after merged active prediction reconfiguration for the testing dataset (Table 7), indicating that an overwhelming majority of ligands predicted as actives were true actives. Based on increases in classification performance metrics as well as the observed hit rate, we feel that the merged active reconfiguration after the initial prediction of ligand function was justified. When analyzing merged active predictions per structure type (experimentally determined ligand–receptor complexes vs. modeled ligand–receptor complexes, Table 8 and Table 9), classifier performance was more accurate for ligand–receptor complexes involving experimentally determined structures (hit rate = 97.6%, accuracy, precision, and recall = 0.96, Table 8) than ligand–receptor complexes involving modeled structures (hit rate = 80.6%, accuracy, precision, and recall = 0.60, Table 9). This result was not surprising, since ligand–receptor complexes involving modeled structures were all generated via ligand docking with structures that exhibited a range of structural accuracies when compared to experimentally determined reference structures (alpha carbon RMSD range = 2.41–5.68 Å, Table 4).

For ligand–receptor complexes in the external dataset, the merged active reconfiguration of initial predictions using majority rule resulted in increases in accuracy, precision, and recall metric values (accuracy, precision, and recall = 0.79, respectively, Table 10) when compared to initial majority rule predictions (accuracy, precision, and recall = 0.04, Table 6) further supporting the notion that merging active classes after making initial ligand function predictions leads to better classifier performance. In addition, the observed binder hit rate of 82.6% (Table 10) indicated that the classifier was again capable of identifying true actives in the sets of ligands it classified as active. Classification of the external dataset using majority rule was also examined per **GPCR model–ligand** pairing, where function prediction for external set ligands docked into in-house [29,30,31], AlphaFold [32,33], GPCRdb [26] active template, and GPCRdb inactive template homology models led to similar observed hit rates (83.3%, 80%, 83.3%, 80%, respectively, Table 11). Overall, the results of this analysis indicated that our classifier was capable of accurately classifying active orthosteric ligands (agonists, antagonists, and inverse agonists) interaction profile fingerprints extracted from binding modes of ligands docked into a variety of homology models.

The machine learning classifier developed in this work is applicable only to predict whether a potential ligand will interact at the orthosteric site of a specific GPCR target. A similar classifier could be developed to predict whether a potential ligand will interact at the allosteric site of a specific GPCR target once a suitable number of GPCR complexes with allosteric ligands becomes available to provide the experimental complexes upon which such a classifier would need to be trained. GPCRdb indicates that only 34 complexes of GPCR with allosteric ligands are currently publicly available, so an allosteric site classifier is not feasible at the current time.

Given that we have developed a classifier that is capable of reliably identifying ligands that may bind to a given GPCR target, we wish to set forth a virtual screening protocol that incorporates our classifier. Using an experimentally determined or modeled structure of a GPCR target, prospective orthosteric ligands should be docked using a method incorporating side chain flexibility. To allow for majority rule predictions of ligand function for each prospective ligand’s set of docked poses, an odd number of docked poses should be retained upon the conclusion of ligand docking. Once docked poses are obtained, interaction energies and types can then be extracted into a dataset that can be fed into our classifier to predict whether a prospective ligand is likely to bind to a GPCR target. The performance of the machine learning classifier described here is challenging to compare to other computational methods utilized in virtual screening workflows, as variable performance between different GPCR targets is likely for each such workflow. However, several virtual screening efforts using TAAR5, which shares a 60% TM similarity with TAAR2, have been published. In 2022, the Atomwise AtomNet^®^ convolutional neural network virtual screening technology was utilized to select a chemically diverse set of 94 predicted hits, of which 2 antagonists with low micromolar potency were identified for a 2% hit rate [265]. In 2023, a virtual screening protocol that included both pharmacophore filtering and docking was used to identify TAAR5 antagonists with a 10% hit rate [266]. Relative to these studies, the machine classifier developed in this study is very promising as a virtual screening tool.

## Figures and Tables

**Figure 1 ijms-25-06876-f001:**
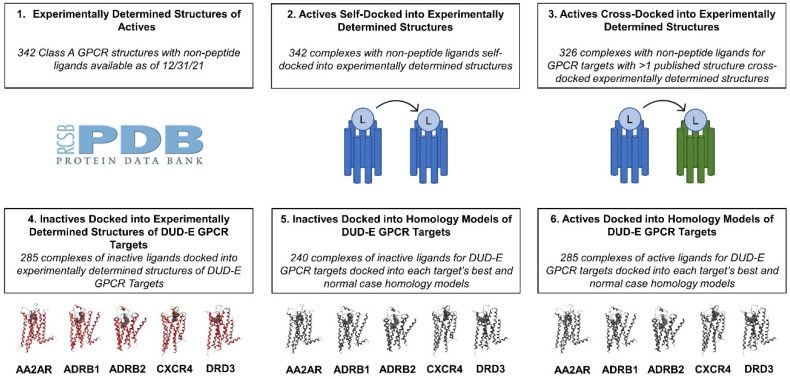
Sources of the six types of ligand–receptor complexes used in our internal dataset. Green and blue cartoon structures are used to represent two different experimentally-determined structures of the same G protein-coupled receptor (GPCR) characterized with the ligand represented by the blue circle labeled ‘L’. Colored ribbon structures represent experimentally-determined GPCR structures and gray ribbon structures represent homology modeled GPCR structures.

**Figure 2 ijms-25-06876-f002:**
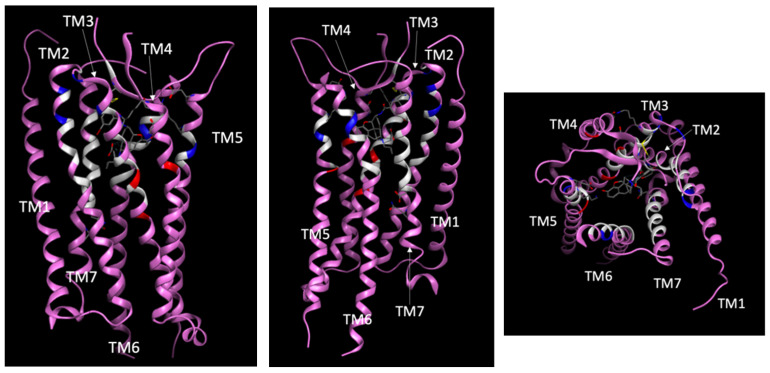
Opioid κ receptor (illustrated using Protein Data Bank (PDB) [23] entry 6B73 [162]) ligand interactions observed in the inactive receptor bound with the antagonist JDTic (PDB entry 4DJH [161]) and active receptor bound with the agonist MP1104 (PDB entry 6B73). Pink ribbons indicate sites lacking ligand interactions. Ribbons color-coded according to the legend indicate sites interacting with ligands in either the active receptor complex (red), the inactive receptor complex (blue), or both receptor complexes (white). TM segments are labeled. The middle image is 180° rotated from the left image. The right image shows the top view (looking down at the receptor from the extracellular side of the membrane).

**Figure 3 ijms-25-06876-f003:**
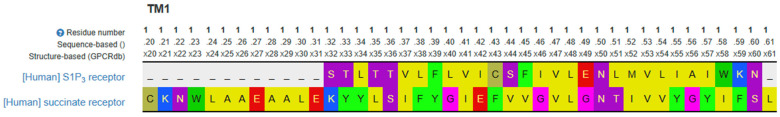
GPCRdb [26] sequence alignment illustrating the differences in transmembrane domain 1 residue numbering for succinate receptor 1 and sphingosine 1-phosphate receptor 3. Yellow, green, purple, red and blue indicate hydrophobic aliphatic, aromatic, polar uncharged, anionic, and cationic amino acid sidechains, respectively. The residue numbers all start with the single-digit numeral in the top row and are followed by either the sequence-based number in the second row or the structure-based number in the third row.

**Figure 4 ijms-25-06876-f004:**
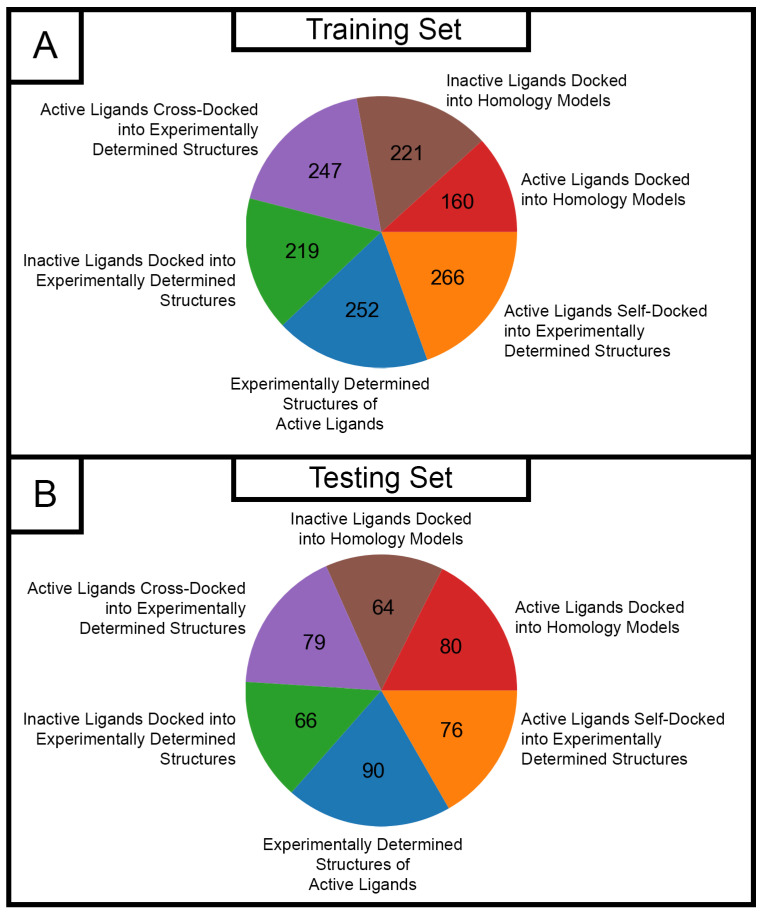
Distributions of ligand–receptor complex types in the (**A**) training and (**B**) testing sets used in classifier development.

**Table 1 ijms-25-06876-t001:** Sites included in ligand interaction fingerprints. Global fingerprints A and B were generated using the 311 ligand complex structures of the 60 receptors shown in Appendix A with no interaction score cutoff (global fingerprint A, included sites exceed 60% weighted interaction percentage) or with a 0.5 interaction score cutoff (global fingerprint B, included sites exceed 40% weighted interaction percentage). The active (included sites exceed 45% weighted interaction percentage), inactive (included sites exceed 40% weighted interaction percentage), and intermediate (included sites exceed 35% weighted interaction percentage) fingerprints were generated using interactions exceeding the 0.5 interaction score cutoff from ligand complex structures of receptors in the named activation state shown in Appendix A.

Global Fingerprint A	Global Fingerprint B	Active Fingerprint	Inactive Fingerprint	Intermediate Fingerprint
2.53				
2.57				
				2.64
				3.24
3.28				3.28
3.29	3.29	3.29	3.29	3.29
3.31				
3.32	3.32	3.32	3.32	
3.33	3.33	3.33	3.33	3.33
3.34				
3.36	3.36	3.36	3.36	
3.37				3.37
45.51	45.51			45.51
45.52	45.52	45.52	45.52	
	6.48	6.48	6.48	
	6.51	6.51	6.51	6.51
	6.52		6.52	
6.55	6.55	6.55	6.55	6.55
				6.58
		7.35		7.35
	7.39	7.39	7.39	
		7.43		

**Table 2 ijms-25-06876-t002:** Summary of cross-docking performance comparing MOE’s SiteFinder function and global ligand fingerprints A and B. Sampling is evaluated as the percentage of complexes with the lowest ligand RMSD (compared to experimental structures) in the noted ranges located within the top 400 scored poses. Scoring is evaluated as the percentage of complexes with the lowest ligand RMSD in the noted ranges located with the top 5 scored poses. Docking results are categorized as successful (<2 Å), acceptable (2–3 Å), successful + acceptable (<3 Å), and unsuccessful (>3 Å) (*n* = 20).

Sampling	Percentage				
Site Selection	Lowest RMSD within	<2 Å (Successful)	2–3 Å (Acceptable)	<3 Å (Successful + acceptable)	>3 Å (Unsuccessful)
SiteFinder	Top 400	30.0	15.0	45.0	55.0
Fingerprint A		35.0	20.0	55.0	45.0
Fingerprint B		35.0	25.0	60.0	40.0
**Scoring**	**Percentage**				
Site Selection	Lowest RMSD within	<2 Å (Successful)	2–3 Å (Acceptable)	<3 Å (Successful + acceptable)	>3 Å (Unsuccessful)
SiteFinder	Top 5	15.0	0.0	15.0	85.0
Fingerprint A		10.0	20.0	30.0	70.0
Fingerprint B		20.0	15.0	35.0	65.0

**Table 3 ijms-25-06876-t003:** Summary of cross-docking performance comparing MOE’s SiteFinder function and receptor state-derived fingerprints. Sampling is evaluated as the percentage of complexes with the lowest ligand RMSD located within the top 400 scored poses. Scoring is evaluated as the percentage of complexes with the lowest ligand RMSD located within the top 5 scored poses. Results are presented as percentages of complexes showing ligand RMSD values (compared to experimental structures) in the successful (<2 Å), acceptable (2–3 Å), successful + acceptable (<3 Å), and unsuccessful (>3 Å) categories (*n* = 8 active; *n* = 16 inactive; *n* = 16 intermediate).

Sampling		Percentage				
Site Selection	Receptor State	Lowest RMSD within	<2 Å (Successful)	2–3 Å (Acceptable)	<3 Å (Successful + acceptable)	>3 Å (Unsuccessful)
SiteFinder	Active	Top 400	62.5	25.0	87.5	12.5
	Inactive		50.0	12.5	62.5	37.5
	Intermediate		68.8	0.0	68.8	31.3
Fingerprint	Active		87.5	0.0	87.5	12.5
	Inactive		56.3	18.8	75.0	25.0
	Intermediate		75.0	0.0	75.0	25.0
**Scoring**		**Percentage**				
Site Selection	Receptor State	Lowest RMSD within	<2 Å (Successful)	2–3 Å (Acceptable)	<3 Å (Successful + acceptable)	>3 Å (Unsuccessful)
SiteFinder	Active	Top 5	0.0	50.0	50.0	50.0
	Inactive		31.3	18.8	50.0	50.0
	Intermediate		43.8	18.8	62.5	37.5
Fingerprint	Active		50.0	0.0	50.0	50.0
	Inactive		31.3	12.5	43.8	56.3
	Intermediate		50.0	12.5	62.5	37.5

**Table 4 ijms-25-06876-t004:** Homology modeling of DUD-E GPCR targets used to construct training and testing datasets.

	Target Receptor	Template Receptor	CoINPocket Score ^c^	Template PDBID	Template State	Reference PDBID ^d^	Reference State ^e^	RMSD (Å) ^f^
Best Case ^a^	AA2AR	AA1AR	4.49	7LD3 [193]	Active	2YDV [55]	Active	3.99
	AA2AR	AA1AR	4.49	5UEN [47]	Inactive	5NM4 [66]	Inactive	4.48
	ADRB1	ADRB2	4.56	4LDE [91]	Active	7BU7 [194]	Active	3.28
	ADRB1	ADRB2	4.56	6PS2 [149]	Inactive	7BVQ [194]	Inactive	3.29
	ADRB2	ADRB1	4.56	7BU7 [194]	Active	4LDE [91]	Active	2.83
	ADRB2	ADRB1	4.56	7BVQ [194]	Inactive	6PS2 [149]	Inactive	5.24
	CXCR4	CCR1	2.37	7VL9 [195]	Active	3ODU [129]	Inactive	5.68
	CXCR4	CCR9	2.21	5LWE [120]	Inactive	3ODU	Inactive	5.25
	DRD3	DRD2	5.08	7JVR [196]	Active	7CMV [197]	Active	2.84
	DRD3	DRD2	5.08	6CM4 [132]	Inactive	3PBL [133]	Inactive	2.71
Normal Case ^b^	AA2AR	S1PR5	1.24	7EW1 [198]	Active	2YDV	Active	5.47
	ADRB1	5HT6	3.40	7XTB [199]	Active	7BU7	Active	2.41
	ADRB2	DRD2	2.80	6LUQ [131]	Inactive	6PS2	Inactive	4.73
	CXCR4	AGTR2	1.72	5UNH [106]	Active	3ODU	Inactive	4.97
	DRD3	5HT1D	3.26	7E32 [200]	Active	7CMV	Active	2.76

^a^ Best-case homology models represent cases where selected template receptors bind to the same endogenous ligand as the target. In addition, best-case homology models represent models constructed using both active and inactive state template structures. ^b^ Normal-case homology models represent cases where a selected template receptor does not bind the same endogenous ligand as the target. ^c^ Maximal self-similarity measure of 5.47. A pairing of two receptors with a local similarity score of 5 would indicate very similar ligand binding pockets, while a receptor pairing with a local similarity score of 1 or less would indicate low ligand binding pocket similarity. ^d^ The Protein Data Bank identification (PDBID) of the reference structure used to compute root-mean-squared deviation values for each modeled structure. ^e^ Activation state of the reference structure used to compute root-mean-squared deviation values for each modeled structure. ^f^ Alpha carbon root-mean-squared deviation value calculated after superposing each homology model onto its target’s experimentally determined structure retrieved from DUD-E.

**Table 5 ijms-25-06876-t005:** Confusion matrix and performance metrics resulting from internal testing dataset classification with the random forest classifier.

Testing Dataset Confusion Matrix					
		Predicted Function			
		Agonist	Antagonist	Inverse Agonist	Inactive
Actual Function	Agonist	94	14	0	21
	Antagonist	16	127	1	35
	Inverse agonist	1	7	9	0
	Inactive	9	4	0	117
**Classifier Performance Metrics**					
Training set cross-validation score				0.80	
Testing set accuracy				0.76	
Testing set precision				0.76	
Testing set recall				0.76	

**Table 6 ijms-25-06876-t006:** Confusion matrix and performance metrics resulting from external dataset classification with the random forest classifier.

External Dataset Confusion Matrices									
Initial Predictions (120 Docked Complexes)					Majority Rule Predictions (24 GPCR Model–Ligand Pairings)				
		Predicted Function					Predicted Function		
		Agonist	Antagonist	Inactive			Agonist	Antagonist	Inactive
Actual Function	Agonist	4	88	8	Actual Function	Agonist	1	18	1
	Inactive	4	16	0		Inactive	0	4	0
**Classifier Performance Metrics**									
**Initial Predictions (120 Docked Complexes)**					**Majority Rule Predictions (24 GPCR Model–Ligand Pairings)**				
Accuracy		0.03			Accuracy		0.04		
Precision		0.03			Precision		0.04		
Recall		0.03			Recall		0.04		

**Table 7 ijms-25-06876-t007:** Confusion matrix and performance metrics resulting from testing dataset classification with the random forest classifier after merging actives.

Testing Dataset Confusion Matrix			
		Predicted Function	
		Active	Inactive
Actual Function	Active	269	56
	Inactive	13	117
**Classifier Performance Metrics**			
Testing set hit rate (%)			95.4
Testing set accuracy			0.85
Testing set precision			0.85
Testing set recall			0.85

**Table 8 ijms-25-06876-t008:** Confusion matrices and performance metrics for merged active predictions (left) and initial predictions (right) after the prediction of ligand function with the random forest classifier for ligand–receptor complexes involving experimentally determined structures.

Testing Dataset Confusion Matrices									
Merged Active Predictions				Initial Predictions					
		Predicted Function				Predicted Function			
		Active	Inactive			Agonist	Antagonist	Inverse Agonist	Inactive
Actual Function	Active	240	5	Actual Function	Agonist	84	9	0	2
					Antagonist	10	119	1	3
	Inactive	6	60		Inverse Agonist	1	7	9	0
					Inactive	5	1	0	60
**Classifier Performance Metrics**									
**Merged Active Predictions**				**Initial Predictions**					
Hit rate (%)		97.6		Hit Rate (%)		NA ^a^			
Accuracy		0.96		Accuracy		0.87			
Precision		0.96		Precision		0.87			
Recall		0.96		Recall		0.87			

^a^ A hit rate was not calculated for non-binary predictions.

**Table 9 ijms-25-06876-t009:** Confusion matrices and performance metrics for merged active predictions (left) and initial predictions (right) after prediction of ligand function with the random forest classifier for ligand–receptor complexes involving modeled structures.

Testing Dataset Confusion Matrices									
Merged Active Predictions					Initial Predictions				
		Predicted Function					Predicted Function		
		Active		Inactive			Agonist	Antagonist	Inactive
Actual Function	Active	29		51	Actual Function	Agonist	10	5	19
						Antagonist	6	8	32
	Inactive	7		57		Inactive	4	3	57
**Classifier Performance Metrics**									
**Merged Active Predictions**					**Initial Predictions**				
Hit rate (%)			80.6		Hit Rate (%)		NA ^a^		
Accuracy			0.60		Accuracy		0.52		
Precision			0.60		Precision		0.52		
Recall			0.60		Recall		0.52		

^a^ A hit rate was not calculated for non-binary predictions.

**Table 10 ijms-25-06876-t010:** Confusion matrix and performance metrics resulting from external dataset classification with the random forest classifier after merging actives and applying majority rule voting per ligand.

External Dataset Confusion Matrix			
		Predicted Function	
		Active	Inactive
Actual Function	Active	19	1
	Inactive	4	0
**Classifier Performance Metrics**			
External set hit rate (%)			82.6
External set accuracy			0.79
External set precision			0.79
External set recall			0.79

**Table 11 ijms-25-06876-t011:** Confusion matrices and binder hit rates resulting from external dataset classification with the random forest classifier after merging actives and applying majority rule voting per ligand for each source of modeled structures used to generate docked ligand–receptor complexes in the external dataset.

In-house Homology Models				AlphaFold Homology Models			
		Predicted Function				Predicted Function	
		Active	Inactive			Active	Inactive
Actual Function	Active	5	0	Actual Function	Active	4	1
	Inactive	1	0		Inactive	1	0
Hit Rate = 83.3%				Hit Rate = 80.0%			
**GPCRdb Active Template Homology Models**				**GPCRdb Inactive Template Homology Models**			
		Predicted Function				Predicted Function	
		Active	Inactive			Active	Inactive
Actual Function	Active	5	0	Actual Function	Active	5	0
	Inactive	1	0		Inactive	1	0
Hit rate = 83.3%				Hit rate = 83.3%			

## Data Availability

The raw data supporting the conclusions of this article will be made available by the authors on request. The classifier developed and evaluated in this work has been made freely available at https://github.com/gszwabowski/GPCR_DB_project, accessed on 23 May 2024.

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
