# Peer review of "G Protein-Coupled Receptor–Ligand Pose and Functional Class Prediction"

_ijms, 2024, doi:10.3390/ijms25136876_

Round 1

Reviewer 1 Report

Comments and Suggestions for Authors  

In the present paper, Szwaboski et al. develop a fingerprint-based classifier that leverages a random-forest implementation to distinguish the activity of potential GPCR ligands based on their binding complexes, provided either by experiments or by docking. The results are moderately interesting, and the method may be helpful in enhancing the currently available tools in virtual screening campaigns. I recommend publication after minor revisions.

·      A fundamental point of the paper is the definition of the fingerprint scheme. The authors should clarify what they mean by: “1) The energetic sum of all interactions occurring at the residue; 2) The categorical type of the most energetically favorable interaction occurring at the residue; 3) The numerical energy of the most energetically favorable interaction occurring at the residue.”

·      The performance of the fingerprint scheme as a binding site definition does not seem as promising as the classifier. I recommend revising this section, and if necessary, removing it from the paper. It may be misleading and difficult for readers to understand.

Comments on the Quality of English Language

The English quality should be improved and the authors should perform a careful check to detect typos.

Reviewer 2 Report

Comments and Suggestions for Authors

This manuscript titled ‘GPCR Ligand Pose and Functional Class Prediction’ by Gregory L. Szwabowski and co-workers focus on two questions, one is the role of ligand interaction fingerprints, two is if a random forest classifier could predict ligand function status. While the question under study is of scientific importance, I have the following major comments that need to be addressed:

1.        The writing of the manuscript is very confusing and not clear. I read through the section 2.1 and 2.2 and still do not understand what the authors try to do in those two sections. Until I read the writing in 2.3 and Figure 2, I finally realize what is done and why. I think the major issue is that in 2.1 and 2.2, the authors write about the question under study, and something has been done, without details of what exactly is done and why do we need that. I think the author assume that I should know something when I read this paper, while I’m a computational chemist and partially work on docking and machine learning, but clearly, I lack the necessary knowledge to fill the gap, thus the confusion. The paragraph between line 161 to 173 is helpful to some degree, but more details about what exact been done and why should we do it, will be greatly appreciated. Also have a flow chart figure like Figure 2 early on may help me understand better.

2.        Following the above points, the method section misses enough details. When I got confused by section 2.1 and 2.2, I go to method section and want to read what exactly be done there, but the corresponding method section still do not have the information I look for. After reading the method section, I still do not understand what exactly have been done and why.

3.        Regarding this 2.1 section, my major concern is:  why do we need perfectly retrieve the PDB ligand pose? Our lab does VLS a lot, when we use Maestro to dock native ligand back into protein, the PDB binding pose is NEVER in the top of the list, and that is fine with us. As long as we see the PDB binding pose in top 30 or so, we will move on to screen for other binders. Our succeed rate of finding a hit is high, even though we didn’t recover the PDB pose in the top rank. Through our communication with Schrodinger support team, they didn’t suggest we reach perfect binding pose neither. So my major question is, is there any data suggest retrieve PDB binding pose can improve the succeed rate of ligand screening later? Also, while the native ligand’s binding pose is accurate, does that means the docked ligand pose is also accurate? There is a possibility that when you overfit toward the known binding pose, the predicted binding pose of a new ligand will be off from real case, because the overfitting. I don’t think there is a way to prove that, because most of the time we don’t have structural information of the new ligand.

4.        GPCR orthosteric binding site study is almost saturated by big lab and industry, people are more divert to allosteric binding site nowadays. Can the authors comment on how this method can be applied to allosteric binding site?

5.        Since this manuscript propose a new application, how does the results from this manuscript compared to other manuscript which didn’t use ligand interaction fingerprints? Are they better or worse?

6.        I extremely like the idea and results of identify functional ligand from just binder, I’m very exciting to see the hit rate of identify true active to be pretty high. However, by carefully looking at table 10, 11, It seems there is a high chance to get false positive, when predict as active, but in reality they are inactive. Can the authors comment on this?

Round 2

Reviewer 2 Report

Comments and Suggestions for Authors

Since my concerns have been addressed. I suggest publishing this manuscript.